# Formic Acid Decomposition Using Palladium-Zinc Preformed Colloidal Nanoparticles Supported on Carbon Nanofibre in Batch and Continuous Flow Reactors: Experimental and Computational Fluid Dynamics Modelling Studies

**DOI:** 10.3390/nano13232993

**Published:** 2023-11-22

**Authors:** Sanaa Hafeez, Eleana Harkou, Panayiota Adamou, Ilaria Barlocco, Elisa Zanella, George Manos, Sultan M. Al-Salem, Xiaowei Chen, Juan Josè Delgado, Nikolaos Dimitratos, Alberto Villa, Achilleas Constantinou

**Affiliations:** 1School of Engineering and Materials Science, Queen Mary University of London, London E1 4NS, UK; sanaa.hafeez@qmul.ac.uk; 2Department of Chemical Engineering, Cyprus University of Technology, 57 Corner of Athinon and Anexartisias, Limassol 3036, Cyprus; ea.harkou@edu.cut.ac.cy (E.H.); paa.adamou@edu.cut.ac.cy (P.A.); 3Department of Chemistry, University of Milan, Via Golgi, 20133 Milan, Italy; ilaria.barlocco@unimib.it (I.B.); elisa.zanella@unimi.it (E.Z.); alberto.villa@unimi.it (A.V.); 4Department of Chemical Engineering, University College London, London WCIE 7JE, UK; g.manos@ucl.ac.uk; 5Environment & Life Sciences Research Centre, Kuwait Institute for Scientific Research, P.O. Box 24885, Safat 13109, Kuwait; ssalem@kisr.edu.kw; 6Department of Materials Science Metallurgical Engineering and Inorganic Chemistry, University of Cádiz, Campus Río San Pedro, E-11510 Puerto Real, Spain; xiaowei.chen@uca.es (X.C.); juanjose.delgado@uca.es (J.J.D.); 7Department of Industrial Chemistry “Toso Montanari”, Alma Mater Studiorum University of Bologna, Viale Risorgimento 4, 40136 Bologna, Italy; nikolaos.dimitratos@unibo.it; 8Center for Chemical Catalysis-C3, Alma Mater Studiorum University of Bologna, Viale Risorgimento 4, 40136 Bologna, Italy

**Keywords:** formic acid, hydrogen, batch, packed-bed flow reactor, CFD

## Abstract

The need to replace conventional fuels with renewable sources is a great challenge for the science community. H_2_ is a promising alternative due to its high energy density and availability. H_2_ generation from formic acid (FA) decomposition occurred in a batch and a packed-bed flow reactor, in mild conditions, using a 2% Pd_6_Zn_4_/HHT (high heated treated) catalyst synthesised via the sol-immobilisation method. Experimental and theoretical studies took place, and the results showed that in the batch system, the conversion was enhanced with increasing reaction temperature, while in the continuous flow system, the conversion was found to decrease due to the deactivation of the catalyst resulting from the generation of the poisoning CO. Computational fluid dynamics (CFD) studies were developed to predict the conversion profiles, which demonstrated great validation with the experimental results. The model can accurately predict the decomposition of FA as well as the deactivation that occurs in the continuous flow system. Of significance was the performance of the packed-bed flow reactor, which showed improved FA conversion in comparison to the batch reactor, potentially leading to the utilisation of continuous flow systems for future fuel cell applications for on-site H_2_ production.

## 1. Introduction

Our dependence on non-renewable energy sources, with their exhaustion in the near future and their effect on the environment, are very crucial factors and challenges that this world needs to consider and face. Hydrogen (H_2_) constitutes an alternative solution, but the storage and distribution of H_2_ remain the most important challenges. Liquid H_2_ is the most desirable form of H_2_ for storage, which involves a complex process and high cost [1]. The storage of H_2_ in other molecules that can be easily transported and utilised is the most dominant option, with formic acid (FA; HCOOH) gaining the interest from scientists as it offers high H_2_ content [2]. Hydrogen fuel cells are considered a promising and a well-established technology for power generation in the future, since they can be utilised in the public transportation sectors, such as in vehicles, submarines and spacecrafts, as well as for portable devices [3,4]. The reaction of the decomposition of FA to H_2_ may operate under mild conditions with the reactant having low toxicity [5], and can be generated through the hydrogenation of atmospheric CO_2_, implying significant positive effects on the climate [6]. FA decomposition follows two routes, with the first one including the production of H_2_ and CO_2_ and the second one the generation of CO and H_2_O. The production of CO is undesirable as it poisons the catalyst, so it must be eliminated [7].

Many studies include investigations of heterogeneous catalysts, as they are most preferable due to the fact that are more easy to use in the separation process with fluid components after the end of a reaction and, most of the time, can be reused [8]. Among the existing investigations, in order to find the optimum heterogeneous catalyst that is highly tolerant to the CO generation pathway but also offers high conversion rates at moderate temperatures, Pd-based catalysts were found to be promising [9]. In recent years, monometallic and bimetallic Pd heterogeneous catalysts were studied for the production of H_2_-rich gas thorugh the decomposition of FA [10,11,12,13,14,15,16,17,18]. Santos et al. [19] synthesised bimetallic PdAu catalysts unsupported and supported on activated carbon with different Pd:Au compositions. Bimetallic catalysts resulted in a smaller particle size compared to monometallic Pd/C and Au/C. Moreover, the supported catalysts were more active than the unsupported catalysts, highlighting the beneficial effect of support. Pd_25_Au_75_/C achieved a TOF value of 212 h^−1^, and the authors concluded that the synergistic effect of the two metals increases the TOF value and enhances the availability of Pd sites on the catalytic surface.

As far as we are aware, Zn metal has been assessed for enhancing the activity of other metals like Pt for FA decomposition [20]. Since Pd-based catalysts are known for being the most active towards H_2_ production from the point of view of FA dehydrogenation [9], and due to the fact that the formation of CO is still a challenge, the investigation of Pd-Zn catalysts with higher activity is important. Zhang et al. [20] synthesised Pd-Zn nanocrystals which that highly stable and showed better resistance to CO poisoning compared to Pd/C. Fathirad et al. [21] investigated the oxidation of FA in a fuel cell utilising Pd-Zn nanoalloys deposited on Vulcan XC-72R carbon. It was found that the bimetallic alloy is a promising candidate for the application of FA fuel cells, as the catalyst exhibited high stability. Hafeez et al. [22] studied FA decomposition using a 2% Pd_6_Zn_4_ catalyst in a stirred batch reactor system, aiming to reproduce their experimental results using process modelling. The comparison between the bimetallic catalyst and the commercial Pd/C catalyst showed that Pd_6_Zn_4_ can achieve higher conversions.

More studies assessing the performance of other bimetallic catalyst have been conducted in order to enhance the activity of monometallic catalysts. Wu et al. [23] reported that the performance of a bimetallic Pd-Co/C catalyst with nonprecious Co metal achieved better catalytic activity and stability than the Pd/C catalyst. In comparison with other Pd-based catalysts with nonprecious metals, the Pd-Co/C catalyst achieved the best activity, attaining a TOF value at 333 K of 8117 h^−1^. It was observed that the presence of Pd^2+^ was higher in the Pd-Co/C catalyst, which may enhance the decomposition of FA, and also, the presence of nonprecious Co improved the cyclability of the catalyst, as the activity of the catalyst after 10 cycles was better. A Pd-Rh-based catalyst was designed at different Pd:Rh ratios by Barlocco et al. [24], as the introduction of a second metal can enhance the properties of the Pd catalyst. It was found that the metal ratio affected the performance of the catalysts, with the Pd-rich catalysts being more active and stable during recycling tests than the Rh-rich ones, with the optimum Pd_90_Rh_10_ catalyst showing activity of 1792 h^−1^.

The decomposition of FA has been performed by many scientists in fixed-bed, stirred-tank reactors and microreactors. Batch reactors are mostly used for the production of small quantities in contrast to fixed-bed reactors that can utilise large-scale capacities. Packed-bed reactors are more preferable as they offer more flexibility in operation, and have the potential to be used as fuel cell systems where the usage of batch reactors is limited in these application systems [25]. Winkler et al. [26] performed an experimental and theoretical investigation on FA dehydrogenation in packed-bed reactors using a Pd/C catalyst. The results disclosed that the catalyst is selective to the reaction towards H_2_ production. A long-term stability test revealed that the catalyst was active even after 8 h. Three kinetic models were applied to describe the experimental data, with the multistep adsorption–decomposition model having the best fit.

Hafeez et al. [27] performed a computational investigation of different microreactor designs with the use of a Pd/C catalyst. The configurations were a packed-bed, a coated-wall and a packed-bed membrane microreactor, with the first two microreactors exhibiting similar performance. The membrane configuration showed better performance compared to the other microreactors, as the separation of the CO improved the conversion of the reactor. A commercial Pd/C catalyst was used for the decomposition of FA by Caiti et al. [28] under mild reaction conditions, revealing that the pore fouling and poisoning induced by formate ions result in catalyst deactivation under continuous operation. Moreover, these factors were found to be enhanced in a plug flow reactor (PFR), as the deactivation was found to be extensive. Minimising the steady state concentration of FA in a continuous flow mode could optimise the results, since the system could operate continuously, under mild conditions, for more than 2500 turnovers, without any loss in activity.

In a previous work reported by our group [22], process simulation modelling was used to investigate and validate the Pd_6_Zn_4_ catalyst in a batch reactor, and we reported better performance of the bimetallic catalyst compared to the Pd/C. In this present work, a Pd_6_Zn_4_/HHT (high heat-treated) catalyst was synthesised through a different preparation method, aiming to improve the overall performance of the catalyst compared to the previous one. The importance of the work was represented by its combination of experimental work with computational simulation in batch and continuous flow reactors, as well as the implementation of the continuous flow system for the constant production of a H_2_ stream for fuel cell applications. CFD modelling studies provide a better understanding of parameter behaviour as it offers the opportunity to study specific terms in the governing equations [11,27,29,30,31,32,33,34].

## 2. Materials and Methods

### 2.1. Catalyst Preparation

The Pd_6_Zn_4_ supported on HHT-CNFs was synthesised via sol-immobilisation in an inert atmosphere for the prevention of Zn oxidation. The morphology and structure of the synthesised catalyst were determined via transmission electron microscopy (TEM) from Thermo Fisher Scientific, Waltham, MA, USA and X-ray photoelectron spectroscopy (XPS) from Thermo Fisher Scientific, Breda, The Netherlands. The details of the materials used, the synthesis of the catalyst and the experimental procedure concerning the characterisation of the catalyst are available in the Appendix A.

### 2.2. Catalytic Tests

The catalytic tests were performed in a batch and a home-made packed-bed reactor where the effects of temperature and FA flow rate were studied, respectively. In both cases, the conversion of FA was evaluated using high-performance liquid chromatography (HPLC). Details of the experimental set up are provided in the Appendix A.

## 3. Modelling Methodology

For this study, COMSOL Multiphysics 5.6 was used in order to couple the mass balance equations and the conservation along with the boundary conditions. All the equations used are presented in the Appendix A. The geometry for the packed-bed microreactor model comprised a mesh containing 641,837 domain elements and 3356 boundary elements, with 77,168 degrees of freedom, while the batch reactor had 3 degrees of freedom. The computational times for the packed-bed and batch systems were 30 s and 13 s, respectively.

## 4. Results and Discussion

### 4.1. Catalyst Characterisation

The Pd_6_Zn_4_/HHT bimetallic catalyst was prepared via the sol-immobilisation method using polyvinyl alcohol (PVA) as a protective agent and NaBH_4_ as a reducing agent [22]. The catalytic performance was evaluated via FA dehydrogenation in a batch and a fixed-bed reactor. TEM was used to evaluate the morphology of the PdZn bimetallic catalyst (Figure 1), showing an average particle size of 2–3 nm, with particles well dispersed on the surface of the carbon nanofibres. STEM-XEDS analysis verified the presence of Pd-Zn bimetallic particles, with an average molar ratio of 72:28, slightly higher than to the nominal one (60:40), due the non-complete immobilisation of Zn on the carbon support, as confirmed via ICP analysis.

### 4.2. Modelling Results

#### 4.2.1. Batch Reactor Validation

An evaluation of the performance of the catalyst was conducted by decomposing FA in a batch reactor. Figure 2 presents the experimental data of the conversion of FA versus time at a temperature range of 30–60 °C compared with the CFD results, which show good agreement. It can be observed that the increase in temperature increases the conversion of FA, and based on the Arrhenius expression (k = A$e^{\frac{-E_{a}}{RT}}$), when the temperature of the reaction is increased, the rate constant, and therefore, the rate of the reaction, are also increased (r = kC_A_). The activation energy for 2% Pd_6_Zn_4_ was found to be around 22 kJ/mol for the batch reactor system. The great overall performance of the catalyst can be attributed to the Pd-enriched surface of the bimetallic catalyst, as confirmed via XPS analysis (Appendix A).

A targeted comparison among the experimental and computational studies using the Pd_6_Zn_4_@HHT catalyst, prepared via the wet impregnation method at 30 °C, is demonstrated in Figure 3. It is evident from the figure that there is a significant difference among the experimental and modelling results compared to Figure 2, since the conversion is lower than the predicted one. This difference might be due to the side reaction generating CO, blocking the active sites of the catalyst while reducing its activity. CO deactivation has not been included in the theoretical investigation. On the contrary, good agreement between the experimental and the modelling results was obtained for Pd_6_Zn_4_@HHT prepared via sol-immobilisation. Indeed, the H_2_ selectivity, measured with the use of an on-line micro-GC for the calculation of CO/CO_2_, was 93% using Pd_6_Zn_4_@HHT prepared via sol-immobilisation, higher than that prepared via wet impregnation (76%). The comparison between the two PdZn catalysts highlights the beneficial effect of the presence of polyvinyl alcohol on the surface of PdZn in terms of selectivity.

#### 4.2.2. Packed-Bed Flow Reactor Validation

The dehydrogenation of FA was also studied in a packed-bed reactor under mild operating conditions. Usually, temperatures above 60 °C are not investigated due to their incapability to be utilised in portable devices like fuel cells for on-site H_2_ generation; therefore, we focused on investigating the reaction temperature only at 30 °C. From Figure 4, it is observed that the conversion reaches up to 50% in the first 1 h of the reaction, followed by a decrease afterwards. The decrease in the conversion is related to the poisoning of the catalyst from the generation of CO due to the dehydration of FA. Moreover, the computational analysis showed great validation, suggesting that a packed-bed flow reactor model is able to predict the reaction profile, including the dehydrogenation of FA to H_2_ generation, as well as the deactivation profile. The continuous flow reactors, due to the fact that they can operate and handle higher concentrations when compared to the batch systems, are more preferred, especially for large-scale processes [25]. Comparing the conversion of FA between the batch and packed-bed reactor systems at 30 °C (Figure 2 and Figure 4), it is found that the latter showed an enhancement in terms of FA conversion.

The designed model includes the solid catalyst, and the internal and external mass transfer limitations within the packed-bed reactor can be determined. The study of mass transfer resistances can display the parameters that lead the reaction to be diffusion- or mass-limited. Figure 5 shows the concentration profile of FA within the catalyst particles. The concentration profiles were obtained at different reactor heights of y = 0.1; 0.25 and 0.35 mm. For the concentration gradient within the catalyst particle, the internal mass transfer resistance is responsible. The CFD models were simulated using constant reactor properties and catalyst particle size. It was observed that there was no significant difference in the concentration of FA (less than 2%), and it can be considered that the internal mass transfer resistance is negligible.

Additionally, mass transfer limitations in the bulk were also investigated. The decomposition of FA involves the diffusion and mass transfer of FA to the area of catalyst particles. The concentration of FA surrounding the catalyst particle was compared to the concentration on the surface of the particle in order to define the external mass transfer limitations. Figure 6 shows a comparison between the bulk concentrations of FA and the concentrations obtained on the surface of the catalyst particle. The difference between the concentration of FA in the bulk and on the surface of the catalyst particle was found to be less than 2%, and it can be considered that the external mass transfer resistance is negligible.

Non-isothermal conditions were also investigated in order to define any heat transfer limitations. Even though the packed-bed reactor operates isothermally at 30 °C, an energy balance was included for this investigation. Figure 7 shows the temperature profile along the length of the bed in non-isothermal conditions. The heat transfer effects are eliminated in the packed-bed reactor as it appears from the temperature profile where the temperature is constant along the bed length.

The inlet flow rate of FA was also investigated (0.1–0.5 mL/min) both experimentally and computationally with a steady reaction temperature of 30 °C. As depicted in Figure 8, lower flow rates resulted in higher conversions. This can be attributed to the dependence on the residence time; the velocity of FA is lower where it can stay longer in the reactor, therefore leading to a higher conversion. Moreover, it was found that 0.1 mL/min had the best performance, achieving FA conversion around 50%, and after 1 h of reaction, the conversion decreased due to deactivation of the catalyst. The activity of the catalyst could be prolonged by the use of membranes since they offer the ability to separate selectively chemical species in the reactor. Since CO is known as a poisonous species in the reaction of FA decomposition, a membrane can be introduced in the reactor to remove the CO that causes deactivation and to alleviate the performance of the catalyst [27].

## 5. Conclusions

Batch and packed-bed configurations were utilised for the decomposition of FA utilising a 2% Pd_6_Zn_4_@HHT catalyst. The studies included the correlation of experimental and CFD studies, which were performed for validation, showing the validity of the experimental results. The results showed that in the batch reactor system, the conversion increases with the reaction temperature, exceeding 80% at 60 °C. The conversion in the continuous flow system was found to be higher at lower inlet FA flow rates, and the activity of the catalyst was significantly decreased after 60 min of reaction time, attributed to the formation of CO, which promotes the poisoning of the active sites of the catalyst. Lastly, switching from a batch reactor to a continuous flow system offers the opportunity to obtain higher FA conversions and potential utilisation for the constant generation of H_2_ in fuel cell applications. The improvement in the activity of the catalyst, since the generated CO promotes its deactivation, is a challenge to overcome in future work. Also, CFD simulations can be utilised for the investigation of different parameters considering the configuration of the reactor.

## Figures and Tables

**Figure 1 nanomaterials-13-02993-f001:**
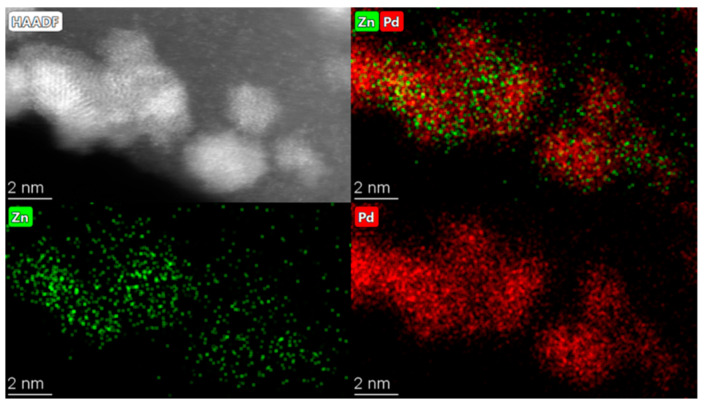
STEM-HAADF image of Pd_6_Zn_4_ catalyst.

**Figure 2 nanomaterials-13-02993-f002:**
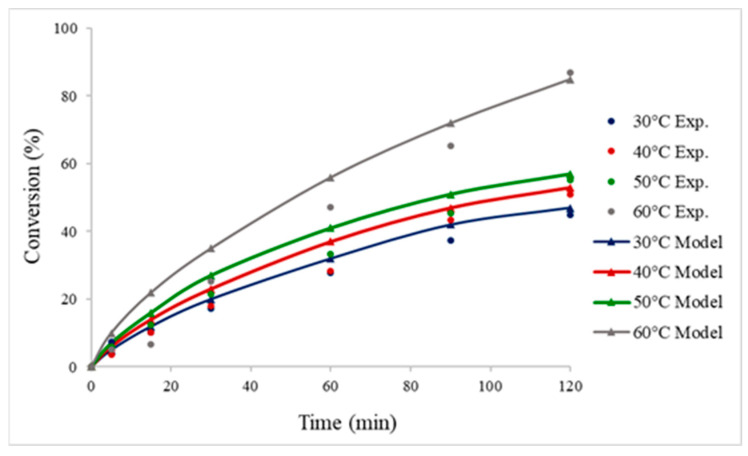
Conversion of formic acid, varying the reaction temperature using 2% Pd_6_Zn_4_@HHT catalyst prepared by sol-immobilisation. Reaction conditions: FA = 0.5 M; FA: metal molar ratio = 2000:1; rpm = 1400.

**Figure 3 nanomaterials-13-02993-f003:**
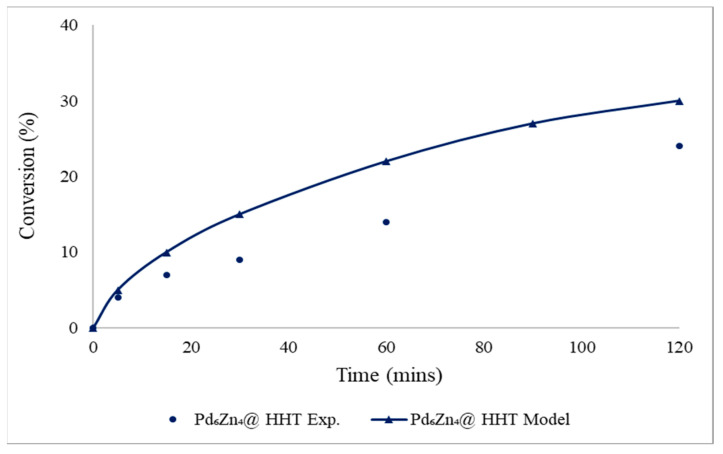
Comparison between modelled and experimental 2% Pd_6_Zn_4_@HHT catalyst using wet impregnation method. Reaction conditions: FA = 0.5 M; FA: metal molar ratio = 2000:1; rpm = 1400.

**Figure 4 nanomaterials-13-02993-f004:**
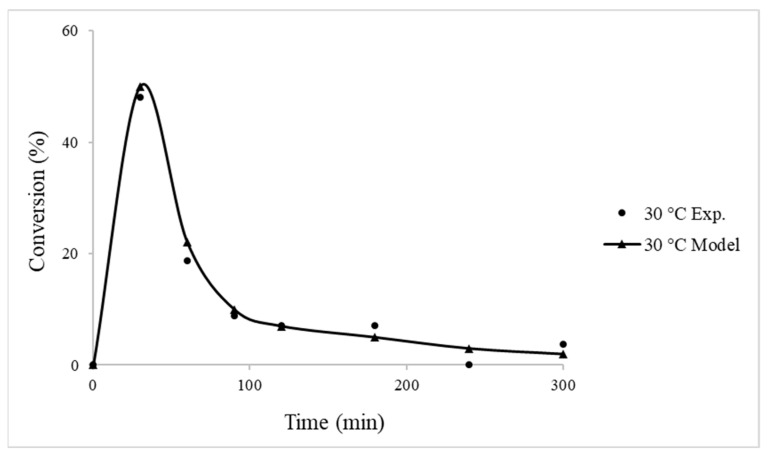
Comparison between experimental and computational modelling data using 2% Pd_6_Zn_4_@HHT catalyst. Reaction conditions: bed length of 7 cm (50 mg Pd_6_Zn_4_/HHT-CNFs and 50 mg HHT-CNFs); FA = 0.1 M; FA flow = 0.1 mL min^−1^; temperature = 30 °C.

**Figure 5 nanomaterials-13-02993-f005:**
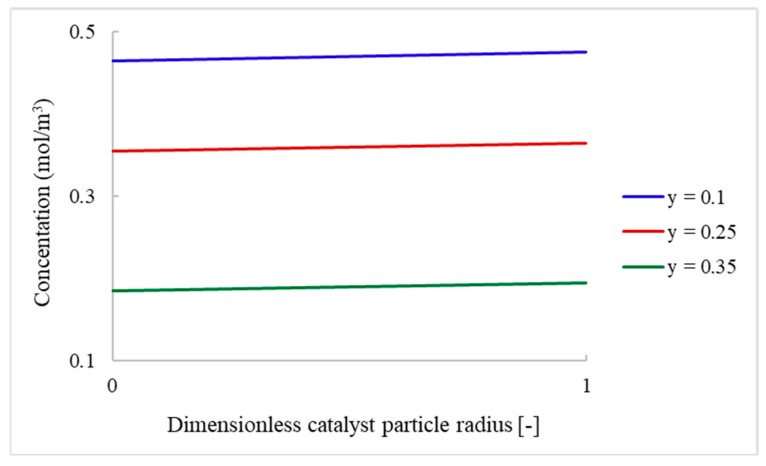
Concentration of FA within the catalyst particles generated from CFD models.

**Figure 6 nanomaterials-13-02993-f006:**
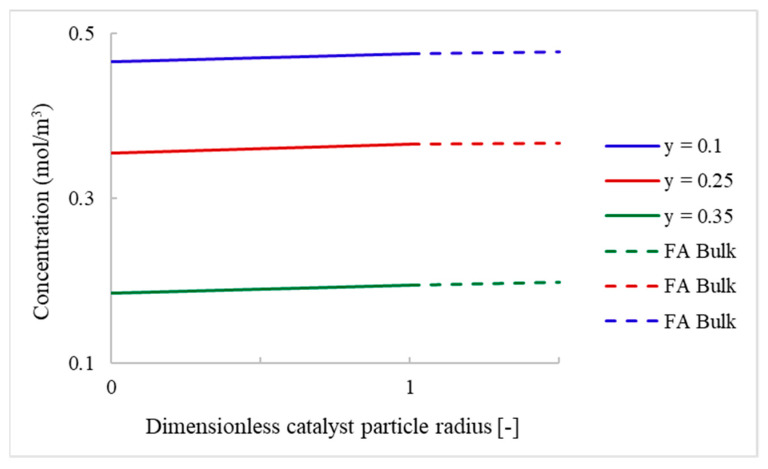
Comparison between the concentration of FA within the catalyst particle and in the bulk generated from the CFD models.

**Figure 7 nanomaterials-13-02993-f007:**
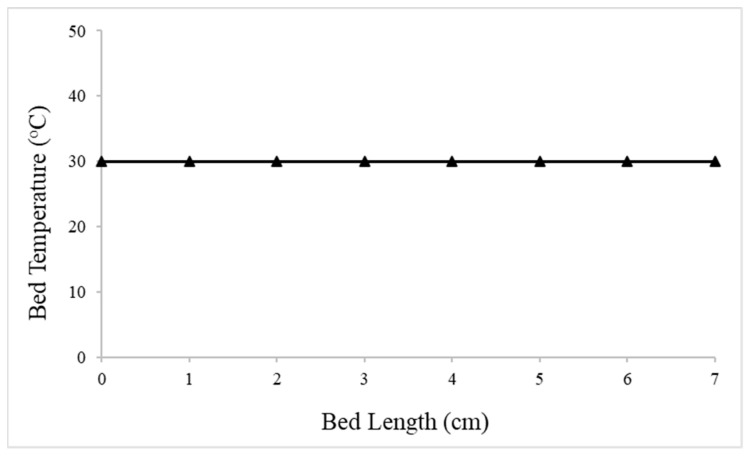
Temperature profile along the length of the bed of the reactor in non-isothermal conditions.

**Figure 8 nanomaterials-13-02993-f008:**
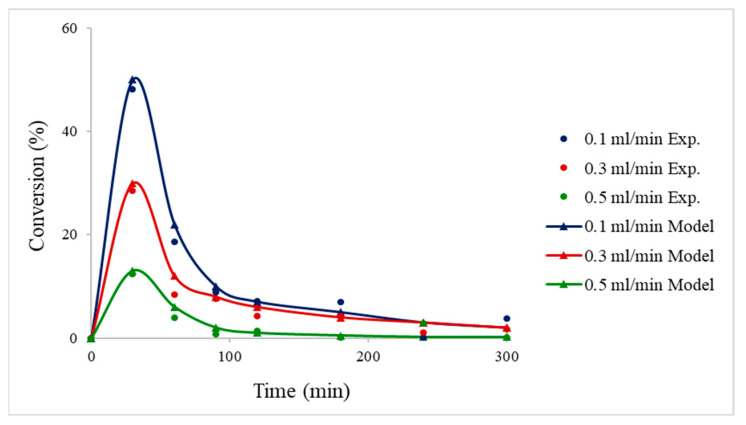
Effects of flow rate on the conversion of formic acid using 2% Pd_6_Zn_4_@HHT catalyst. Reaction conditions: bed length of 7 cm (50 mg Pd_6_Zn_4_/HHT-CNFs and 50 mg HHT-CNFs); FA = 0.1 M; FA flow = 0.1, 0.3, 0.5 mL min^−1^; temperature = 30 °C.

## Data Availability

The data presented in this study are available from the authors.

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
