# Peer review of "Formic Acid Decomposition Using Palladium-Zinc Preformed Colloidal Nanoparticles Supported on Carbon Nanofibre in Batch and Continuous Flow Reactors: Experimental and Computational Fluid Dynamics Modelling Studies"

_nanomaterials, 2023, doi:10.3390/nano13232993_

Round 1
Reviewer 1 Report
Comments and Suggestions for Authors
This paper denotes characterization of the Pd/Zn bimetallic catalyst for the formic acid-to- H2 conversion in batch and continuous flow reactors. The manuscript was written persuasively. Particularly, the section of Introduction was well demonstrated with technical present condition. Minor revision is required.
1. In the Abstract, write "HHT" in complete words.
2. Figure numbering seems to be wrong in the text. For example, there is no demonstration for Figure 2. All check again.
Author Response
Please see rebuttal document.

Reviewer 2 Report
Comments and Suggestions for Authors
This paper investigates a simulation and experimental study of the decomposition of formic acid to produce hydrogen using palladium-zinc preformed colloidal nanoparticles supported on carbon nanofibres.
However, there are a number of issues that need to be clarified.
1. Supplementary material. Page 3, Equation (3). Why is the power-law model used to represent reaction rates? For solid catalytic reactions, why not use the Langmuir-Hinshelwood kinetic model?
2. Supplementary material. Page 3, Section 2.1. The reactor model in the main manuscript does not have any intermittent reactor diagrams. This is difficult to understand. For example, does the hydrogen flow out of the batch reactor or does it stay in the batch reactor(the pressure will increase in the reactor?)? How is the heat of reaction removed using the cooling jacket?
3. Supplementary material. Page 3, Section 2.1. There are no analogue diagrams in the batch reactor. Is the temperature distribution in the reactor uniform?
4. Supplementary material. Page 4, Section 2.3. The PBR reactor model does not contain any reactor diagrams. For example, does hydrogen flow out of the PBR reactor? How does the heat of reaction remove using the cooling jacket?
5. Supplementary material. Page 4, Section 2.3. There are no analogue diagrams in the PBR reactor. What is the temperature profile(axis and radius direction) in the PBR reactor?
6. Supplementary material. Page 4, Section 2.3. The PBR reactor is a three-phase reactor (liquid(reactant,FA)-Gas(H2, CO, CO2)-Solid(catalyst)). Does there exist a two phase flow in the reactor effluent? Three is no any description about the gas flow model in your PBR reactor? How does your PBR reactor experiment overcome this problem?
7. Please add more CFD simulation result of the PBR and Batch reactor in the main manuscript.
I suggest major revisions to the manuscript.
Author Response
Please see rebuttal document

Reviewer 3 Report
Comments and Suggestions for Authors
The manuscript discusses the generation of H2 from formic acid (FA) decomposition in a batch and a packed bed flow reactor, under mild conditions, using a 2% Pd6Zn4/HHT catalyst synthesized by sol-immobilization method. The experimental results were well supported by CFD modeling studies. The presented work is related to previous work reported by the same group where process simulation modeling was used to investigate and validate the Pd6Zn4 catalyst in a batch reactor, reporting the better performance of the bimetallic catalyst compared to the Pd/C. The novelty and added value of this work is well elaborated and sufficiently justified.
However, there are a few shortcomings that could be mentioned. Firstly, Figure 2 is poorly connected to the manuscript text. Secondly, “figure 2a” is mentioned in Supplementary material. It should be “Figure S1 a”. Thirdly, it is unclear whether the quality of H2 received is pure enough to be used for example in PEM. Finally, as the activity of the catalyst was rather short (about 60 minutes), it would be useful to mention how it could be prolonged.
Author Response
Please see rebuttal document
